# Listening to parents to understand their priorities for autism research

**Megan Clark** [1,2] *, **Dawn Adams** [1,2]

**1** Autism Centre of Excellence, School of Education and Professional Studies, Griffith University, Brisbane, Australia, **2** Griffith Institute for Educational Research, Griffith University, Brisbane, Australia

* Megan.Clark@Griffith.edu.au

**Data Availability Statement:** All relevant data are within the paper and its Supporting Information files.

**Funding:** Support from the Griffith University Partnership with the Autism Community to Enhance Research (PACER) grant.

## Abstract

Involving the autism community in research increases the real-world translation and impact of findings. The current study explored the research priorities of parents of school-aged children on the autism spectrum across the home, school, and community settings. A combination of content analysis of an online questionnaire ($n = 134$) and Q-sort methodology ($n = 9$) was used. The most commonly identified research priorities in the online questionnaire were child *health and well-being* (home setting), *socialisation and social support* (school), and *community awareness and understanding of autism* (community). The Q-sort method highlighted different top priorities, with *understanding the parent*, *sibling, child and family impact and stress* the highest ranked priority for home, *teacher/staff education and support* for the school, and *recognizing and supporting anxiety* for the community. The implications of the findings are discussed in relation to shifting the framework of autism research to align research agendas with parental priorities.

## Introduction

Autism spectrum disorders (ASD; hereafter autism) are a series of lifelong neurodevelopmental conditions that currently affect one in 59 children [1]. Differences or difficulties with social communication and restricted, repetitive behaviours and patterns of interest are characteristic of autism [2]. An increase in the diagnostic prevalence of autism has seen a subsequent increase in autism research over the past decade [3]. As the volume of autism research continues to increase, researchers should consider how to engage the autism community in the conceptualisation of ideas, to promote involvement in each stage of the research cycle. The autism community comprises individuals living with autism and their families, allied health professionals, and educators. Collectively, these individuals form an important group of stakeholders with a lived or shared experience of autism, and valuable insight that can inform the direction that future research should take.

Researchers are beginning to understand the importance of engaging members of the autism community (including individuals on the autism spectrum, their family members, educators and allied health professionals working in the field) in setting priorities. The new era of autism research seeks to engage the autism community meaningfully, valuing the collaboration

**Competing interests:** The authors have declared that no competing interests exist.

between researchers and community members, so that the needs and experiences of individuals on the autism spectrum and their families, can inform the research priorities and agenda for research [4]. Community engagement in autism research can increase the translation of science and improve the application of findings in practice, thereby working towards reducing the research to practice gap [5,6]

Research involves many phases from the conceptualisation of ideas, establishment of a project and recruitment of participants, to the dissemination of findings. However, research engagement often occurs partway through the process, often during the recruitment phase, or following the completion of a project when findings are disseminated [7]. However, by this stage of the research process, the ideas have been established, the project has been developed and the study is already underway or completed, with decisions often pre-determined and driven by researchers' agendas. This leaves little room for the input from non-researchers to shape the direction of the study, and unsurprisingly, this has left community members feeling dissatisfied and even frustrated with their limited involvement in research [5].

Engagement can involve participation in any stage of the research process, from the establishment of research agendas through to the dissemination of findings [7]. Thus, it is important that researchers adopt a more inclusive research ethos and make an effort to engage the community earlier in the research process. One way of meaningfully engaging the autism community in research from the outset is through priority setting, allowing the experiences, ideas and needs of community members to influence the development of a project from the outset.

## Priority setting

Priority setting is a framework that involves collecting and synthesising the views, knowledge, and experience of stakeholders to guide future decision-making [8]. It is useful in identifying the range of needs within a target population, so that research can be directed towards those areas where it is most needed to maximize the impact. Although such frameworks are becoming increasingly integral to research funders, they should be seen not simply as a funding body requirement, but rather as a meaningful and beneficial process of ensuring that research is both relevant and useful outside of academia [9]. Research priorities of the autism community can be established using a number of methods including an online questionnaire, interviews, and focus groups, with online questionnaires one of the most common methods used. Research priority setting has previously been completed with some members of the autism community, but for the most part, these studies have focused upon identifying priorities for the early intervention years [10–13]. These studies sought to determine parent's priorities for early intervention, including prioritizing goals for the early intervention years in preparation for school. However, as these studies focused on younger children prior to school entry, it cannot be assumed that the needs and priorities identified by parents during the early years will remain priorities as children transition into school. Thus, it is important that the specific priorities most relevant to the school years are identified, so that research can be targeted to the areas of most importance to support children during this developmental stage. As needs can change at various stages of development, needs prioritisation should occur frequently and at different life stages, so supports can be tailored to individuals on the spectrum across the lifespan [14].

A large-scale survey was conducted in the UK to determine if autism research funding aligned with the research priorities of the autism community [5]. A total of 1,517 responders completed the survey, including autistic adults ($n$ = 122), immediate family members (including parents, children and siblings of individuals on the spectrum of any age; $n$ = 849), professionals ($n$ = 426), and autism researchers ($n$ = 120). Findings indicated a disparity between the

allocation of research funding and the priorities articulated by the stakeholders. Funding patterns revealed that research exploring the risk factors (37%), diagnosis (13%), and biological basis of autism (18%) received the greatest funding, which did not coincide with the community's priorities and preference for more research that focused on improving the lives of individuals living with autism. While family members wanted to see more research into transitions (i.e., to school and employment) and research promoting safety and independence for children on the spectrum, research on lifespan issues was the least well-funded area, indicating an incongruity between the areas that have been most commonly researched and the priorities and needs of the autism community.

This pattern of spending is not specific to the UK; a recent analysis of Australian funding revealed a similar pattern of spending, with 47% of research funding during 2008–2012 being allocated to biological research, and treatments or interventions receiving only 22% [15]. Even after the introduction of the Cooperative Research Centre for Living with Autism (Autism CRC), the world's first national, cooperative research effort focused on autism which added almost AUD$20,000,000 to autism research in Australia, biological research continued to receive the greatest amount of funding. Similar to the UK, this suggests that research was failing to target the priorities of the autism community within Australia.

The majority of studies seeking to establish the priorites for the autism community have been conducted internationally, predominantly across the US and UK [16]. A largescale study explored the priorities of the autism community within a diverse group of stakeholders (*n* = 6,004) including persons with autism, family members, researchers, clinicians/educators, and others from America (86.4%), Canada (10.2%), and the UK (2%) [17]. Four research priorities for future autism research were identified by all groups: understanding co-occurring conditions, adult transitions, health and well-being, and lifespan issues. Although these were clearly identified as priorities across this large sample, many of these areas have been identified as under-researched, again highlighting the discrepancy between the priorities identified and the research currently being undertaken. For example, although anxiety is recognized as one of the most common co-occurring conditions for children on the spectrum [18], and has been shown to have a significant impact on the educational quality of life of children on the spectrum [19], a systematic review on anxiety in children on the autism spectrum notes a lack of research in this area [20]. Using such large and diverse samples has a number of benefits, but it may also result in general community priorities rather than those which can benefit specific subsections of the autism community, such as older adults, adults or school-aged children.

## The current study

Exploration of the autism research funding in the UK and in Australian identified that autism research within Australia has failed to align with the priorities of the autism community [21]. The autism community has an integral role to play in establishing research priorities to ensure that autism research is more responsive to the immediate needs of autistic people and their families [5]. Research that is informed by priorities of the community can increase its relevance and translation into the community, as long as the priorities are tailored to specific groups of the community. Despite the prevalence of autism being highest amongst school-aged children [22], to date there has not been any research priority setting that is specific to this cohort. Parents are often the greatest advocates for their child and thus, are especially important stakeholders when prioritising the needs of research for school-aged children. However, the research priorities of parents of school-aged children are largely understudied and consequently, the understanding of what the research priorities should be for school-aged children on the autism spectrum is currently limited. Such knowledge will help to guide the

allocation of research funding and ensure that research is being done where it is needed, and thus, is most likely to have a real impact across all aspects (home, school, community) of the lives of individuals on the autism spectrum, and their families.

## Aims

This study adds to the small but growing body of research detailing research priorities by being the first to focus specifically upon research priorities to benefit school-aged children with autism. Drawing upon the literature focusing upon participation of individuals on the autism spectrum [23,24], parents were asked to consider their priorities for research to support their child in a range of settings; the child's home, the child's school and the child when out in the community. The following research questions were posed:

What do parents identify as priorities for autism research to support their school-aged children across the following settings: (a) home, (b) school, and (c) community?

In addition, informal examination of parent discussions was undertaken to explore:

1. why some priorities were identified as more important than others in the Q-sort ranking, and

2. the factors that parents identify as important for their decision-making around research priorities.

## Method

### Procedure

Parents' research priorities were established and explored across two phases: an online questionnaire, followed by Q-sort ranking and group discussions. Consent was obtained from the Griffith University Human Ethics Board, approval number: GU_2018_888.

**Phase 1: Online questionnaire.** An online questionnaire was advertised via social media and shared with community contacts including autism organisations, parent groups, and autism research groups both nationally and internationally. Parents were first presented with a plain language statement outlining the nature of the study and were advised that, if they proceeded with the questionnaire, their consent was implied. Participation was voluntary, and the questionnaire took approximately 20 minutes to complete. If parents had more than one child on the spectrum in the family, they were asked to base their priorities on only one of their children at a time, and were provided the opportunity to complete the questionnaire a second time to comment on their priorities for their other child(ren). Parents were asked the following question to ascertain their priorities for autism research: "In your opinion what three areas should research focus on to support your child on the autism spectrum in each of the following settings: home, school and community?" A minimum of one priority for each setting was required, but parents could provide as many as three priorities per setting. When providing multiple responses for each setting, parents were asked to list their priorities in order of importance from most to least important. This phase asked parents about their research priorities across three settings (home, school and community) to obtain a cross-contextual perspective of where parents perceive research is needed to support children on the autism spectrum across different environments. A sample of the questionnaire items can be found in the S1 File.

**Phase 2: Q-sort.** Q-sort methodology was implemented in the second phase of the study to gain further insight into parents' research priorities gathered from the online questionnaire

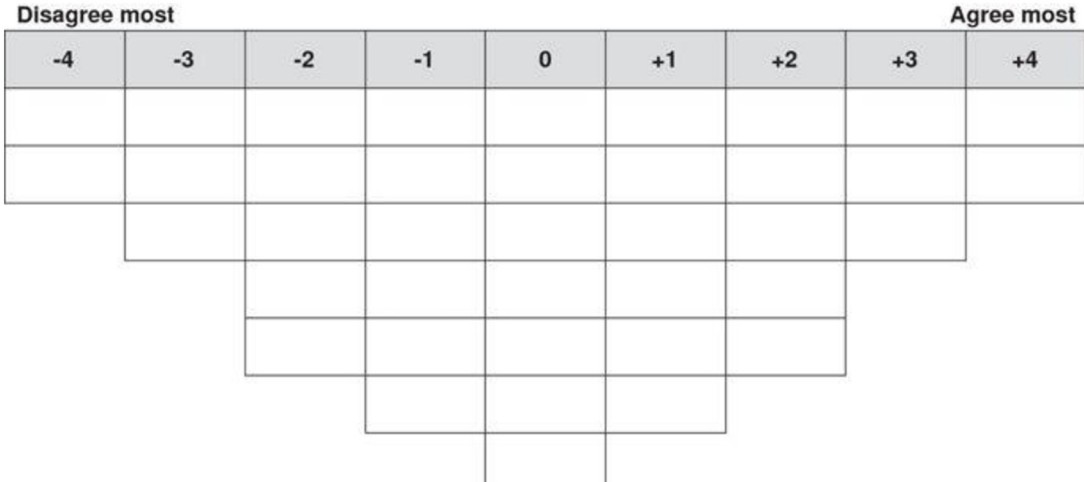

**Fig 1. Q-sort template used to arrange priorities from most to least important.**

during phase 1. Q-methodology is a technique that combines qualitative and quantitative methods to understand participants' viewpoints on a range of topics [25] and has been widely used to identify priorities for research, healthcare, and policies [26–30]. As per the work of [26] only the "forced sort" Q-sort task (and not the factor analysis) of Q-methodology was used in the current study. The top 15 categories of priorities for home and school, and all 13 research priority categories for the community were derived from the online questionnaire responses. Each priority was individually printed onto cards for parents to arrange from most important (+4) through to least important (-4) using the Q-sort grid (see Fig 1).

Parents were invited to attend the research centre in groups of three. Upon arrival at the centre, parents were provided with an information statement outlining the purpose of the research and were given the opportunity to ask questions before providing written consent. Prior to commencing the Q-sort task, researchers explained to parents' that the role of this exercise was to further explore the top priorities for autism research that parents had identified in an online questionnaire. Parents' were first shown the Q-sort grid printed on A3 paper and were provided with the following instructions: "In this activity you will arrange the top 15 priorities gathered from parents in an online questionnaire to determine where research is most needed to support school-aged children on the autism spectrum at [home, school community]. Each priority is printed on a separate card and you will be using this grid to rank your priorities from most important at this end, to least important at this end. You will have 10 minutes to work independently and arrange your priorities. Once you are happy with your ranking you will use the blue-tak to secure each priority onto their respective square on this grid". Parents arranged priorities for one setting at a time and the order of settings (home, school, community) was counterbalanced across the three focus groups (i.e., in Group 1, home was presented first while either school or community was presented first in the subsequent groups). Once all participants had completed their sort, they were then asked to discuss and explore the reasons behind their choices and rankings within their groups. The discussions were recorded on an audio-recording device.

## Participants

**Phase 1: Online questionnaire.** One hundred and thirty-four parents (98.5% mothers) of school-aged children on the spectrum completed the online questionnaire. Parents were aged

29–62, with a $M_{age}$ of 42 years ($SD$ = 6.12), with the majority of parents (97%) residing in Australia, four of the parents completed the questionnaire from outside of Australia.

**Phase 2: Q-sort.** Nine parents of school-aged children on the spectrum participated in one of three Q-sort and group discussion sessions; each group was comprised of three parents. Groups 1 and 3 had secondary-school-aged children while parents in Group 2 had younger children at primary school. The majority of parents involved in the groups fell within the age bracket of 41 to 50 years (66.7%).

The demographics of the parents involved in both research phases are presented in Table 1.

## Data analysis

**Phase 1: Online questionnaire.** The qualitative data for the open-answer questions were analysed by the first author and researcher using content analysis, a method for identifying, analysing, and reporting patterns or themes within data. Content analysis has been an established method of coding qualitative data in autism research for more than 30 years [31–33]. Coding involved four-steps: (1) data were divided into manageable parts, (2) responses related to the areas or questions of interest were collated, (3) categories were created that described similar responses, and (4) categories were combined or split where data could best be described by a rearranged structure [34]. A 5% cut-off rule was also adopted whereby a minimum number of responses within each category were required for that category to be retained. For example, where 75 responses were coded, each category required a minimum of four responses [35,36]. The second author independently coded 20% of the responses across each setting that were selected at random into categories to assess reliability. Inter-rater agreement was calculated using Kappa. Agreement was reached when both raters coded a given answer into the same category. Kappa for the first round of inter-rater reliability was .94 for priorities in the home, .97 for priorities at school, and .95 for community research priorities, all of which indicate "near perfect" agreement. Raters met for further discussion and clarification of category names where the final inter-rater agreement reached 100% across all settings.

Categories were created across the three priorities within each setting so that ratings could be combined to create a single research priorities list for home, school, and community. Where more than one priority was provided within each allocated response box, only the first response was coded. Each category required a minimum of five responses across the first, second, and third priorities to be retained. Once all the data were coded into categories, the category which was identified by the highest number of parents was given a ranking of 1, the category which was listed by the next highest number of parents was given a ranking of 2, and so on.

**Table 1. Demographics of Parents within Phase 1 and Phase 2.**

| | | Phase 1: Online questionnaire | | Phase 2: Q-sort | |
|---|---|---|---|---|---|
| | | *n* | *%* | *n* | *%* |
| Parent gender | Male | 2 | (1.5%) | 1 | (11.1%) |
| | Female | 132 | (98.5%) | 8 | (88.9%) |
| Number of children in the family | 1–3 children | 116 | (86.57%) | 7 | (77.7%) |
| | 4–7 children | 18 | (13.43%) | 2 | (22.3%) |
| Number of children diagnosed with autism in family | One child | 104 | (77.62%) | 5 | (55.5%) |
| | Two or more children | 30 | (22.38%) | 4 | (44.5%) |
| Parent education | No formal education | 1 | (0.74%) | 0 | (0%) |
| | Completed secondary school | 15 | (11.2%) | 3 | (33.3%) |
| | Completed tertiary education | 118 | (88%) | 6 | (66.7%) |

**Phase 2: Q-sort.** Scores were allocated to each priority dependent upon its placement in the template (Fig 1). First, the mean total rank score for each priority was calculated across the total sample. This was then used to assign a rank for the priorities, with 1 being the priority with the highest rank score and 15 for home and school or 13 for community being the priority with the lowest mean rank score. This was then repeated individually for parents of primary and secondary school-aged children. The division of primary and secondary school groups for data analysis is presented in Table 2. Comments from the discussions were not formally analysed but informally reviewed in order to address the second research question.

## Results

### Priorities for research to support children on the autism spectrum in the home

**Phase 1: Online questionnaire.** Parents were invited to list up to three priorities for research to support their child at home; all 134 parents provided at least one priority, 119 parents (88.8%) provided at least two priorities (i.e., 15 parents only reported one research priority for home), and 103 parents (76.8%) provided three research priorities. For research in the home, parent responses were coded into 23 categories. Research to support children's *health and well-being* was the most commonly identified research priority. The top 15 research priorities from the questionnaire can be found in Table 3, with the second column indicating their ranking based on the online questionnaire results.

**Phase 2: Q-sort.** The top 15 priorities for research in the home were provided to parents to be ranked using Q-sort in Phase 2; for results see columns 3, 4, and 5 of Table 3. The category with the highest Q-sort score across all parents was *parent, sibling, child and family impact and stress*; when explored by school type (primary, secondary), this remained the most important priority for parents of primary school-aged children and was the second most important priority for parents of secondary school children.

Reasons for placing this as the priority became apparent during Q-sort discussion. Parents discussed their concerns about the impact of autism on the other family members, including siblings: "I need intervention and resources to support his siblings, there are four kids in the family, and while he is receiving the support he needs because of his autism. . .I need the support for the rest of us, because our lives are focused in around him". Parents also identified that more work is needed to support others in the family, in addition to the child living with autism.

### Priorities for research to support children on the autism spectrum at school

**Phase 1: Online questionnaire.** In total, 132 parents (98.5%) indicated one priority, 124 parents (92.5%) listed two priorities, and 118 parents (88%) listed the maximum three priorities for future research to support their child in the school setting. Parent responses were coded into 21 categories. The category *socialisation and social support* was the category identified by the highest number of parents as a priority for school-based research.

**Table 2. Primary and secondary school groupings for analyses.**

|  | Primary school | Secondary school |
|---|---|---|
| Total *n* | 5 | 4 |
| Mean age (SD) of children | 9 years (2.54) | 14 years (.57) |
| Age range of children | 6–12 years | 13–17 years |

**Table 3. Priority rankings (and mean Q-sort score) from Phase 1 and Phase 2 for research in the home from the online questionnaire and Q-sort method (sorted in order of highest to lowest priority from the online questionnaire).**

| Priority | Online questionnaire ranking | Ranking (Mean Q-sort score) All participants | Ranking (Mean Q-sort score) Primary school parents | Ranking (Mean Q-sort score) Secondary school parents |
|---|---|---|---|---|
| Health and well-being | 1 | 5 (+2) | 8 (+1.4) | 3 (+2.75) |
| Parent, sibling, child and family support | 2 | 3 (+2.3) | 4 (+1.8) | 1 (+3) |
| Self-care, daily living skills and independence | 3 | 2 (+2.4) | 3 (+2.2) | 4 (+2.75) |
| Behaviour and behaviour support | 4 | 15 (+0.3) | 13 (+0.4) | 14 (+0.25) |
| Emotions, regulation and behaviour | 5 | 6 (+1.7) | 7 (+1.6) | 6 (+2) |
| Parent strategies, training, intervention and resources | 6 | 9 (+1.3) | 10 (+1.2) | 8 (+1.5) |
| Family relationships and interactions | 7 | 13 (+0.6) | 14 (+0.2) | 9 (+1.25) |
| Managing rigidity, routines and supporting change | 8 | 10 (+1.1) | 11 (+0.6) | 7 (+1.75) |
| Intervention and resources for the home | 9 | 12 (+0.7) | 12 (+0.6) | 13 (+1) |
| Speech language and communication | 10 | 8 (+1.5) | 6 (+1.8) | 10 (+1.25) |
| Relationships (social and romantic) | 11 | 14 (+0.6) | 15 (+0.2) | 11 (+1.25) |
| Other diagnoses (i.e., ADHD, ID) | 12 | 4 (+2.2) | 2 (+2.2) | 5 (+2.25) |
| Parent, sibling, child and family impact and stress | 13 | 1 (+3) | 1 (+3.2) | 2 (+2.75) |
| Technology use | 14 | 11 (+0.7) | 5 (+1.8) | 15 (-0.5) |
| Meltdowns and tantrums | 15 | 7 (+1.2) | 9 (+1.4) | 12 (+1) |

**Phase 2: Q-sort.** As shown in Table 4, on the basis of Q-sort rankings, *teacher/staff education and support* was the highest ranked priority across the total sample, which had been ranked fourth in the online survey. When explored by school type (primary, secondary), *teacher/staff education and support* remained the most endorsed priority for parents of primary school-aged children but parents of children at secondary school ranked *inclusive education* as the most important area for research.

During the discussions following the Q-sort, parents collectively expressed concern regarding teachers' knowledge and education of autism, and whether they had the skills and experience to support the needs of their children: "I need the staff to have more of an understanding of his differences". One parent went on to say that having a teacher who understood that her child was different from other children and could only learn through "moving rather than sitting in a chair" made all the difference. Another parent spoke about the value of a teacher who "just got it" and understood that their child's assessment needed to take place in a less conventional manner: "when he was assessed in the classroom he failed the assessment . . . his next assessment was completed while sitting in a tree using a white-board, he passed everything".

### Priorities for research to support children on the autism spectrum when out in the community

**Phase 1: Online questionnaire.** One hundred and twenty-one parents (90.3%) provided one priority for research to support their child on the spectrum within the community, 106 parents (79%) provided two of three possible priorities, and 94 parents (70%) provided three of three possible priorities that were coded into 13 final categories. Research to increase

**Table 4. Priority rankings (and mean Q-sort score) from Phase 1 and Phase 2 for research in the school setting from the online questionnaire and Q-sort method (sorted in order of highest to lowest priority from the online questionnaire).**

| Priority | Online questionnaire ranking | Ranking (Mean Q-sort score) All Participants | Ranking (Mean Q-sort score) Primary school parents | Ranking (Mean Q-sort score) Secondary school parents |
|---|---|---|---|---|
| Socialisation and social support | 1 | 11 (+1.4) | 9 (+2) | 15 (+0.75) |
| School resources and support | 2 | 12 (+1.4) | 13 (+1.2) | 9 (+1.75) |
| Supporting diverse and individual learning needs | 3 | 8 (+2) | 6 (+2.2) | 8 (+1.75) |
| Teacher/staff education and support | 4 | 1 (+3.1) | 1 (+3.2) | 3 (+3) |
| Awareness, understanding and acceptance of autism | 5 | 5 (+2.2) | 3 (+2.4) | 5 (+2) |
| Recognising and supporting anxiety at school | 6 | 2 (+2.6) | 5 (+2.2) | 2 (+3.25) |
| Learning and academic outcomes | 7 | 3 (+1.4) | 11 (+1.8) | 14 (+1) |
| Emotions and regulation | 8 | 13 (+1.6) | 10 (+1.8) | 10 (+1.5) |
| Behaviour and behaviour support | 9 | 10 (+1.1) | 14 (+0.8) | 11 (+1.5) |
| Executive functioning | 10 | 4 (+2.4) | 2 (+2.8) | 6 (+2) |
| Inclusive education | 11 | 3 (+2.6) | 8 (+2) | 1 (+3.5) |
| Curriculum modifications and support | 12 | 6 (+2.1) | 4 (+2.4) | 7 (+1.75) |
| Student participation and engagement | 13 | 7 (+2.1) | 12 (+1.8) | 4 (+2.5) |
| Education and acceptance from my child's peer | 14 | 9 (+1.7) | 11 (+2.2) | 12 (+1.25) |
| Health and well-being | 15 | 15 (+0.7) | 15 (+0.4) | 13 (+1.25) |

Figures in parentheses represent mean Q-sort ranking for each priority.

community awareness, understanding, and acceptance of autism was the most frequently identified research priority.

**Phase 2: Q-sort.** As shown in Table 5, for research in the community, using the Q-sort method, *recognising and supporting anxiety* was identified as the most important priority for autism research across all groups. When explored by school type, *recognising and supporting anxiety* remained the second most important priority of parents in both primary and secondary school groups. The highest rated priority for primary school parents was *community awareness and understanding of autism*, which was also identified as the most important priority on the basis of the online questionnaire in Phase 1. Parents of secondary school-aged children rated *confidence and support for my child to live an independent life* as their highest priority for autism research within the community.

During the discussions after the Q-sort, a number of parents shared their experiences of stigma within their communities, with several similar experiences reflected by that articulated by one parent: "Unfortunately I think it is still the judgement and prejudice that we regularly experience in our community that is stopping my child from moving forward and participating". Based on such experiences, it is easy to understand why *community awareness and understanding of autism* was consistently rated as an important research priority to support children within community settings. One of the parent groups who rated awareness and acceptance of autism within the community identified this priority as a roadblock that needed to be addressed first and foremost before any other research could move forward: "there is no point researching any of these other areas [remaining priorities], until community awareness and

**Table 5. Priority rankings (and Mean Q-sort score) from Phase 1 and Phase 2 for research in the community from the online questionnaire and Q-sort method (sorted in order of highest to lowest priority from the online questionnaire).**

| Priority | Online questionnaire ranking | Ranking (Mean Q-sort score) All participants | Ranking (Mean Q-sort score) Primary school parents | Ranking (Mean Q-sort score) Secondary school parents |
|---|---|---|---|---|
| Community awareness and understanding of autism | 1 | 3 (+2.4) | 1 (+2.8) | 9 (+2) |
| Relationships (social and romantic) and support | 2 | 4 (+2.3) | 3 (+2.4) | 8 (+2.25) |
| Inclusive activities and attitudes | 3 | 11 (+1.7) | 9 (+1.4) | 7 (+2.25) |
| Diagnosis, intervention and support | 4 | 7 (+2) | 11 (+1.4) | 5 (+2.75) |
| Community attitudes | 5 | 6 (+2.1) | 7 (+1.8) | 6 (+2.5) |
| Employment, post-school opportunities and support | 6 | 9 (+2) | 12 (+1.4) | 4 (+2.75) |
| Confidence and support for my child to live an independent life | 7 | 2 (+2.6) | 4 (+2.2) | 1 (+3.25) |
| Community adaptations for inclusive environments and events | 8 | 5 (+2.2) | 8 (+1.6) | 3 (+3) |
| Autism-specific knowledge, education and resources in the community | 9 | 8 (+2) | 6 (+2) | 10 (+2) |
| Recognising and supporting anxiety | 10 | 1 (+2.7) | 2 (+2.4) | 2 (+3) |
| Developing a safer community | 11 | 13 (+0.7) | 13 (+0.6) | 13 (+1) |
| Emotions, regulation and behavior | 12 | 10 (+1.8) | 5 (+2.2) | 11 (+1.5) |
| Help and resources for families | 13 | 12 (+1.4) | 10 (+1.4) | 12 (+1.5) |

Figures in parentheses represent mean Q-sort ranking for each priority.

understanding of autism has been addressed . . . community attitudes won't change until the awareness and understanding of autism has increased".

### Parent-identified factors that influenced their decisions around research priorities

The online questionnaire was able to establish which priorities were most important to parents; however, this method did not capture *why* these priorities were important. The post Q-sort discussions provided parents with a forum to discuss why some priorities were more important than others, and provided valuable insight into their decision-making process that could not be obtained from the online questionnaire. A discussion between parents when deliberating rankings of priorities emphasized the importance of engaging parents in the research process, as it is their lived experiences that can inform researchers and health professionals (and funding bodies) where the support is most needed: "as parents, we know that a lot of these areas are important and that work needs to be done, but do the people who make decisions about the research and funding know this?. . . because these are the people that really need to know".

More often than not, the discussions amongst parents revealed that when priorities were ranked as "less important", this was because they were areas that had already been researched and parents did not necessarily want these to be the focus of more research in the future: "if we look at these in terms of future research, we know there is already loads out there, so that's why these are less important". However, it is also important to recognize that while a priority may not have been rated within the top five priorities within a setting, this is not to say that parents did not find this priority important. This was captured by one parent during the Q-sort activity: "this is hard . . . how am I meant to prioritise some of these over the rest?. . . I can

see that these are all important . . . I guess some are just more important to my family at the moment than others". This reflects the complex nature of the activity that required parents to reach a decision with regards to what was "most important" for research to focus on.

It is also important to consider that the decisions made during the Q-sort task reflect what is happening in the lives of the parents and children at the time of the study. Therefore, if this activity were to be repeated with the same parents in 12 months' time, there is the possibility that the needs of the families and their children may have changed, and this may subsequently affect what priorities are deemed to be most important to parents at that point in time: "don't get me wrong, I think these are all important, but right now, this is the biggest issue that our family is facing"; "if you had of asked me six months ago, I think my answer would have been different".

## Discussion

Autism research is entering a new era that values the involvement of the autism community in the entire research process: from the evolution as ideas are shaped, through data collection, and out into the dissemination, translation, and application of findings. Akin to the previous research priorities within the autism community, the focus was upon supporting individuals and their families in the current moment and not upon research into biological aspects of autism, despite that being the most funded research area in both the UK [5] and Australia [15]. This paper adds to the priority-setting literature by being the first to (i) focus specifically upon the experiences and priorities of parents of primary and secondary school-aged children on the spectrum (ii) combine an online questionnaire and follow-up Q-sort groups to explore priorities and (iii) ask about the research priorities for supporting school-aged children across a range of settings (home, school, community).

### Priorities for research to support children on the autism spectrum at home

Research to support children's *health and well-being* was the highest priority for research in the home from the online questionnaire results. This is an unsurprising finding given co-occurring physical and psychological conditions are frequently reported for children on the spectrum, with almost 40% of children on the spectrum experiencing between one and four co-occurring conditions, and 53.9% of children impacted by four or more co-occurring conditions compared to 11.5% and 0.3% of children without disabilities, respectively [37]. Some of the most frequently reported co-occurring conditions which affect health and well-being have been shown to impact upon not only the child's, but also the parent's, quality of life [38,39], a factor which may be contributing to the desire for more research in this area.

When ranked using the Q-sort method, research into *parent, sibling, child, and family impact and stress* was ranked as the most important research priority overall (with it being the highest rated for primary school parents and second highest rated for parents of children at secondary school). The experience of autism within the family is an area that has received considerable research interest. For example, the increased family stress and mental health difficulties experienced by parents raising a child on the autism spectrum have been well documented [40–42]. Most recently, the impact of autism on other siblings in the family has been explored, with adolescent siblings of individuals on the spectrum self-reporting higher levels of stress than siblings of individuals with Down Syndrome [43]. Interventions are often cantered on supporting the individual living with autism; however, parents in the current study expressed a desire for interventions to support the health and well-being of the other children in their families, whose needs often come second to the needs of their children on the spectrum: "there are other children in the family but often it is the one with the highest needs [child on the

spectrum] that is the focus". While the social-emotional well-being and behavioural adjustment of siblings of children on the autism spectrum have been widely reported [44], there are fewer support options available for siblings. Sibling support groups are one approach that is available to address the needs and to educate siblings on autism while supporting them in their role as a sibling simultaneously [45–48]. Yet, despite the availability of sibling support groups over the past 28 years, there is still little research available to determine the impact that such groups have on the well-being and quality of life of siblings of children on the spectrum [49]. Thus, there is a pressing need for more research to explore interventions and resources that can support the family unit, especially the siblings of children on the spectrum.

## Priorities for research to support children on the autism spectrum at school

*Socialisation and social support* was the most important research priority for the school setting as determined by the online questionnaire. The identification of this as a research priority is interesting given that within the last few years, systematic reviews have been published on social skills interventions [50], school-based interventions to target social communication behaviours [51], and peer-mediated interventions to enhance social behaviour [52]. Despite social needs being well researched, a needs analysis of 101 Canadian families of school-aged children on the spectrum found that social needs were the most common need, with 78% of parents reporting a preference for more supported social activities or programs both in school and extra-curricular (i.e., within the community) to support their child's social development and friendships, and to promote social inclusion for their children [53]. This may suggest a delay or difficulty in translating this research into practice, difficulties in accessing such interventions, and/or that the implications or interventions coming out of the research are not meeting the needs that parents identify in their children on the spectrum in the school setting. Parents in the current study revealed that the social difficulties at school remain a pertinent issue, where one mother reflected on her child's experience by saying, "one of the biggest issues we have faced is friendship difficulties, with other children not being understanding or inclusive of my child".

Parents also wanted to see research to increase *teacher/staff education and support*, with this prioritised in the mean Q-sort (across all three parent groups) and by primary school parents specifically. Collectively, parents believed that if teachers had more knowledge about autism they would be better equipped to accommodate their child's needs at school: "if teachers had more support and training, they would understand that it's not about comparing our kids to neurotypical kids . . . they would understand that autism means a different learning style". More autism-specific education and training for staff appears to be an important research priority both in the eyes of parents and teachers themselves. For example, in a Swedish study, only 14% of staff had received any formal training on educating students with neurodevelopmental disabilities [54]. With regards to ongoing professional development, school principals and teachers have been found to report a lack of professional development opportunities and training in the experience and understanding of autism [55]. Further, a recent study found that less than half of teachers in the study (40.2%) had access to professional development specifically related to supporting their students on the spectrum in their schools [56], suggesting there is a persistent need for research to prioritise autism-specific learning opportunities for staff who are educating students on the spectrum. Research needs to first explore teachers' preferences for autism specific professional development that can inform targeted learning, to equip staff educating students on the autism spectrum.

For parents of secondary school-aged children, *inclusive education* was the greatest priority. While educators generally hold positive attitudes towards the inclusion of students on the spectrum in mainstream classrooms [57,58], and despite the availability of guidelines to assist teachers educating students on the spectrum in mainstream settings [59], such regulations are slower to be translated into real-world practice in the school setting [60]. As many as 70% of children on the spectrum are now attending mainstream classrooms [61], with an estimated 80% of these students requiring additional support in their school setting [62], understandably, parents want to see research that prioritises *inclusive education*, with concerns that their children are not being supported within their school environment: "often there is a societal drive to place children in mainstream schools, even though the system isn't supporting the approach".

Only one Australian study to date reports on the school-based provisions of students on the autism spectrum [56], resulting in a limited understanding of what inclusive education looks like, and specifically how teachers are educating and supporting students on the autism spectrum in the classroom environment. Therefore, it is evident that more research is needed to respond to parents' concerns surrounding inclusive education, to understand how to build a supportive inclusive environment that can best suit the learning needs and maximise the well-being of students on the autism spectrum.

## Priorities for research to support children on the autism spectrum when out in the community

Anxiety was identified as an issue amongst parents overall, with *recognising and supporting anxiety* ranked highest in the total Q-sort ranking across all three groups. Children on the autism spectrum are six times more likely to receive a diagnosis of anxiety when compared to typically developing children [18]. However, recognizing anxiety in children with autism is complicated and this is in part due to what is referred to as "diagnostic overshadowing" caused by the similarities of symptomatology across the two conditions [63]. Parents discussed the complexities in recognizing anxiety in their own children, with a mother of both a daughter and a son with autism discussing the differences in symptom presentation of her two children: "anxiety was easier to pick up in my son because you could see changes in his behaviour and emotions but my daughter is much better at masking her anxiety. . . in boys it is easier to know what to look for, but it is hard to know what anxiety looks like in girls with autism . . . so I think more research is needed to specifically understand anxiety in general, but particularly in girls with autism". If anxiety is not recognised and diagnosed appropriately, children cannot receive the support they need, which poses the risk that children's anxiety will continue to impact their lives. This is especially concerning given the most recent evidence that anxiety is contributing to poorer quality of life for children with autism [19,38]. One parent elaborated on the impact of their child's anxiety on completing necessary tasks in the community such as grocery shopping "It is difficult for us to be in a busy crowded supermarket. . . sometimes I have had to leave without finishing my shopping" and "having to wait in queues at the shops is too difficult for my child and he becomes anxious". In a recent study exploring barriers and enablers impacting children's management of anxiety, *noise, crowds and overstimulation* was the most frequently identified barrier hindering children on the spectrum from managing their own anxiety when in the community [36]. The need for research into ways of recognising and then supporting anxiety when the child is out in community settings or at community events is therefore an important research priority to support community inclusion and participation.

When explored across parents of older and younger children, the priority with the highest Q-sort score by parents of primary school-aged children was research on *community awareness and understanding of autism*, which was also identified as the highest ranked priority from the online questionnaire. Autism acceptance can be defined as being or feeling accepted or appreciated as a person with autism, and/or autism being recognised positively and accepted by others and the self as an integral part of that individual [64]. Both individuals with autism as well as their family members report elevated levels of stigma [65] and unconscious bias has been shown to impact neurotypical adults, including those who work regularly with children with autism [66]. As such, stigma can impact upon the mental health and well-being of the individual with autism as well as their family members [64]. Taken together, the findings suggest that there is a need for research to move beyond describing the presence of stigma and towards intervention to reduce stigma and increase autism acceptance and awareness.

## Methodology matters

A recent systematic review of priority setting studies in autism [16] identified seven published studies, all of which used different methodologies, recruitment methods and methods to set the priorities, suggesting that there is no set standard way for this work to be undertaken. The results from both the online questionnaire and the Q-sort should be interpreted in light of the different methods used to collect data, each relying on different cognitive processes. For example, the open-ended nature of the online questionnaire relied on parent recall, such that parents were required to think about and report on anything of importance to them at the time the questionnaire was completed. In contrast, the Q-sort activity tapped into recognition where parents' priorities were determined from a pre-set list of responses derived from the online questionnaire. This may explain why some of the priorities were weighted differently across the two studies. For example, *behaviour and behaviour support* was rated as the fourth highest priority in the online questionnaire but was considered less important in the Q-sort. It may be that when parents were generating ideas from scratch in the online questionnaire, they were commenting on the most salient issues such as behavioural difficulties at home. However, when parents were presented with a pre-set list of priorities in the Q-sort, this may have included priorities parents had not previously considered to be important based on their current circumstances, but that parents may have rated as an important priority for their child's future.

## Limitations and future research directions

This study is not without limitations. It was not possible to capture children's research priorities in the current study, presenting an important area for future research as such knowledge would help with the co-development of future research projects in the field. Parents' responses in the online survey differed in length, including, on occasion, one-word responses. Whilst parents would understand the context of this answer, as standalone responses submitted in a written format, they could sometimes be ambiguous and would have benefited from further explanation. In most instances, the response itself, albeit one worded, was clear, but too broad to direct specific research. For example, "anxiety" emerged as a recurring response across settings, suggesting that parents are wanting to see this as a focus of research in the home, at school, and in the community. However, when only one word is written, the nature of the research desired here is unclear as anxiety is a broad and complex construct. The one-word response of "anxiety" under a school priority would have been coded under the umbrella category "Ways to recognise and support anxiety at school", but the parents may have had a specific aspect in mind; for example, how anxiety presents at school, how teachers recognised it,

or perhaps how anxiety is impacting on academic performance or adjustment to school or their child's social relationships.

It is acknowledged that the research priorities are not all the same in terms of scope; for example, some areas, such as anxiety, received enough specific parent responses to have its own specific category, whereas other areas were less frequently listed and so were combined to make broader categories. Anxiety was also identified as a separate, independent priority from broader mental health and emotional well-being topics in a priority-setting exercise with stakeholders for children and young people with neurodisability [67], highlighting the importance of this topic within current clinical and research settings [68]. Although the open-ended nature of the questions was a strength of this study, as it provided parents with the opportunity to share their ideas without restrictions, this may have also been a limitation. Without a list of potential priorities to guide them, some parents may default to what is most obvious to them. Future research using open-ended questionnaire formats may benefit from providing parents with some prompting or structure to guide how to approach the open nature questions without influencing their responses. For example, providing parents with the following instructions to set up the questions may provide them with more structure "What questions do you think research should try to answer in order to support children on the autism spectrum in the following settings? *(These questions could begin with what does. . .? Why do. . ..? How can we. . ..? When is. . ..? Or many other combinations)"* Further, encouraging parents to provide a brief explanation or example alongside their response may have yielded more context for one-word responses such as 'anxiety' that could have been interpreted in a number of ways (i.e., understanding the presentation of anxiety, diagnosing, or supporting anxiety).

The umbrella terms derived from the coding, and used to label each of the categories of priorities provided to parents in the Q-sort, may have limited some parents' understanding of the priority areas, and this may explain some of the discrepancy between the online survey and Q-sort rankings. However, verbatim examples from the online survey were provided to parents, ensuring a compatibility between the data from the online survey and the interpretation of terminology in the Q-sort.

It is important to acknowledge the lack of diversity in the sample with the majority of parents (97%) residing within Australia. There is the possibility that the parent priorities may be influenced by cultural and social systems within a country, therefore, future research would benefit from comparison of priorities of parents or stakeholders across different countries. Further, mothers formed the large majority of respondents in both phases of this study. It is not unusual for research to have a larger representation of mothers, than fathers, with mothers often acting as proxy respondents for fathers with work commitments [69]. Despite the challenges associated with recruiting and retaining fathers in research, fathers can add valuable insight into a study [69]. It is unclear whether the priorities identified in the online questionnaire would have remained similar or differed with the inclusion of more fathers, and future priority setting studies should seek to recruit a more diverse sample of parents, with more fathers where possible. Previous priority setting studies have reported on samples as small as 43 [70] and 62 [68]. However, it is acknowledged that the sample size of the online sample was modest and as the Q-sort was only conducted with 9 parents (eight mothers and one father), it is difficult to generalize the findings; an important element to consider when conducting future priority setting work. Nonetheless, this is an important first step in seeking to understand the priorities of parents of school-aged children on the spectrum.

While this study began to explore parents' perceptions of why certain priorities were more important than others, and has included some of the discussion from parents, this was limited to anecdotal comments from parents and did not involve formal analysis of their responses. Future replication of the Q-sort methodology with a larger and more diverse group of parents

(including more fathers), would benefit from use of thematic analysis for coding parent responses to gain further understanding into what influenced parents' decision during the Q-sort activity.

## Implications

The current findings have important implications for the future of autism research and the delivery of services offered to individuals living with autism and their families. Public engagement in research has been identified as one of the key facilitators in the translation of research into practice [71]. Yet, the disconnect between the research agenda and the needs, values, and priorities of the autism community has been well established, both with Australia, and internationally [5,15,16,21] indicating that research is not meeting the needs and preferences of the autism community and, thus, is not having its intended impact. Therefore, establishing the priorities of parents in the current study offers many positive implications to advance the impact of autism research moving forward, informing research agendas, directing policy, impacting attitudes, and influencing grant funding outcomes. While ensuring that priorities can shape the research, this is only part of the equation, as the research prioritised here will have little benefit to the community if this knowledge is not put into action. At the conclusion of the group session, parents were asked with whom the research priorities should be shared. All parents identified that it was important to share the findings with policy and government bodies, including the department of education, with one parent stating they felt that "the education department need to see these findings so they know what is important to parents". It is important that these priorities are disseminated to encourage more research where it is needed. As stated by one mother during her group's discussion, "without research like this that explains what the autism community needs, we are never going to be able to make real change".

The study also has implications for those working across the home, school, and community settings. These research priorities may indicate the settings that parents feel are not getting the best out of their child. It may, therefore, be useful for schools and community groups to explore the priorities listed for schools and to use these to begin to inform their professional development training. It is also useful for professionals, as it may inform them of the topics that are pertinent to parents, and that would be good topics around which to have helpful, quality resources.

## Conclusions

This study has provided insight into parents' priorities for future research to support their child with autism at home, at school, and in the community. Further, the addition of the Q-sort and the discussion task has increased understanding of why these priorities are important to parents. All parents involved in the groups identified that their decision-making was influenced by two factors: the areas of priority represent issues of importance in the life of their child, and they are the topics that parents want to see as the focus of future autism research. Priority setting proved a valuable exercise in delineating the specific areas where autism research should be prioritised to improve the lives of school-aged children with autism. It is important that the priorities identified in the current study guide future decision-making around the development and funding of future research to increase the meaning, impact and relevance of autism research.

## Supporting information

**S1 File. Sample questionnaire.**
(DOCX)

## Acknowledgments

We would like to thank all the parents who gave their time to support this research study.

## Author Contributions

**Conceptualization:** Megan Clark, Dawn Adams.

**Data curation:** Megan Clark.

**Formal analysis:** Megan Clark, Dawn Adams.

**Investigation:** Megan Clark.

**Methodology:** Megan Clark, Dawn Adams.

**Project administration:** Megan Clark.

**Writing – original draft:** Megan Clark.

**Writing – review & editing:** Dawn Adams.

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
