## [Decision Letter · Decision Letter 0]

27 Apr 2020

PONE-D-20-01529

Listening to parents to understand their priorities for autism research

PLOS ONE

Dear Dr Clark,

Thank you for submitting your manuscript to PLOS ONE. This is an important topic, and after careful consideration, by two experts in the field, and my self, we feel that it has merit but does not fully meet PLOS ONE’s publication criteria as it currently stands. Therefore, we invite you to submit a revised version of the manuscript that addresses the points raised during the review process.

In particular, you will see that both reviewers have requested additional details. As you consider these points, please be aware that while PLOS ONE does consider qualitative and mixed-methods studies, we recommend that authors use the COREQ checklist, or other relevant checklists listed by the Equator Network, such as the SRQR, to ensure complete reporting (http://journals.plos.org/plosone/s/submission-guidelines#loc-qualitative-research). In general, we would expect qualitative studies to include the following: 1) defined objectives or research questions; 2) description of the sampling strategy, including rationale for the recruitment method, participant inclusion/exclusion criteria and the number of participants recruited; 3) detailed reporting of the data collection procedures; 4) data analysis procedures described in sufficient detail to enable replication; 5) a discussion of potential sources of bias; and 6) a discussion of limitations. While some of these points have already been addressed in your manuscript, please ensure that all are sufficiently covered as you respond to the reviewers' comments, and utilize the checklists noted above, or some other process to facilitate complete reporting. 

We would appreciate receiving your revised manuscript by Jun 11 2020 11:59PM. To enhance the reproducibility of your results, we recommend that if applicable you deposit your laboratory protocols in protocols.io, where a protocol can be assigned its own identifier (DOI) such that it can be cited independently in the future. For instructions see: http://journals.plos.org/plosone/s/submission-guidelines#loc-laboratory-protocols

We look forward to receiving your revised manuscript.

Kind regards,

Eric J. Moody, Ph.D.

Academic Editor

PLOS ONE

Journal Requirements:

3. Please provide additional details regarding participant consent. In the ethics statement in the Methods and online submission information, please ensure that you have specified, regarding the focus groups part of the study, (a) whether consent was suitably informed and (b) what type you obtained (for instance, written or verbal). If your study included minors under age 18, state whether you obtained consent from parents or guardians. If the need for consent was waived by the ethics committee, please include this information.

"We also acknowledge the support of <removed for blind review> University’s Partnering with the Autism Community to Enhance Research (PACER) grant."

"The authors receive no specific funding for this work"

5. Please include your tables as part of your main manuscript and remove the individual files. Please note that supplementary tables (should remain/ be uploaded) as separate "supporting information" files

Reviewers' comments:

Reviewer's Responses to Questions

**Comments to the Author**

1. Is the manuscript technically sound, and do the data support the conclusions?

Reviewer #1: Yes

Reviewer #2: Partly

2. Has the statistical analysis been performed appropriately and rigorously? 

Reviewer #1: Yes

Reviewer #2: No

3. Have the authors made all data underlying the findings in their manuscript fully available?

Reviewer #1: Yes

Reviewer #2: Yes

4. Is the manuscript presented in an intelligible fashion and written in standard English?

Reviewer #1: No

Reviewer #2: Yes

5. Review Comments to the Author

Reviewer #1: Upon my review, I recommend the manuscript titled: “Listening to parents to understand their priorities for autism research” for a decision of Major Revision for publication in PLOS ONE. Overall, I find that this manuscript offers a strong contribution to the larger area of Autism research. It provides insight into the process of involving parent’s perspectives in research priority setting as well as an analysis of what some of these priorities are within the Autism community. I do however think that the arguments and organization of the manuscript need to be made stronger. Particularly, the overall justification and set up of the manuscript are unclear. While the results and discussion of the data are strong, I found myself coming to my own conclusion about the importance of this work, whereas the manuscript should have made this clearer from the start. I also think that the balance of a discussion of the process of involving parent’s perspectives and the actual data collected in for the manuscript should be considered further. I do find this work to be important to the field and important to share. I have provided some detailed comments in an attached document. The comments are in approximate chronological order in relation to the manuscript. I hope you find these comments helpful.

Reviewer #2: Thank you for the opportunity to review the manuscript titled Listening to parents to understand their priorities for autism research. Priority research is vital to determine the targets most important to the consumers who would benefit most from the research. Considering that most grants awarded focus on biological factors, even though that does not appear to be the most desired area of research for consumers, studies similar to what the authors did in this manuscript need to occur more frequently and should be more widely disseminated. In doing this, we may be able to change the trajectory of autism research. The manuscript was well written and has potential of being a valuable contribution to the field. I have the following suggestions, comments, and questions

Introduction

• Page 4-description of respondents to UK survey-use of word “autistic individuals”; in some circles, this is preferred while in others, person-first language is required for publication. Not sure of the PONE guidelines. This use of “autistic people, etc.” is seen in other sections of the manuscript along with person-first language; thus, its use is inconsistent.

• Page 5-the last sentence before the Current Study section appears to be incomplete.

Method

Procedure

• Were both phases using qualitative procedures? The questionnaire, as described later in the analysis section, appeared to be open-ended.

• Would Q-sort be a “focus group”? Q-sort appears to be an approach to capture both qualitative and quantitative information whereas focus group activities is a separate approach that relies on facilitation of rich discourse between participants in a group with analysis primarily arriving at a coding schema to identify themes. It may be more accurate to state Q-sort approach was used with a subset of the participants who received the questionnaire. The participants in the Q-sort did not appear to have a discussion; rather, they primarily sorted cards that reflected their ideas about priorities. It also may be more accurate to describe it as Q methodology. Although you had discussions with parents after the Q sort was completed, the coding and primary analysis seemed to focus on the Q sort and comments related to priorities. That is, there did not seem to be a qualitative research plan for identifying themes for the parent-identified factors that influenced their decisions in the Q-sort.

Questionnaire

• Please provide more information about the on-line questionnaire. How many questions were included? Or did the questionnaire only include the one question about priorities? I am assuming that it included only one question; however, it is not clear if that is the case.

Q Sort

• How was the Q-sort facilitated? There were 3 groups, each consisting of 3 parents. Describe how the Q-sort approach was conducted for each group and how you ensured consistency of facilitation across all 3 groups.

Participants

• Do you know the total N of parents with children who have autism in the country/area from which you recruited? That is, what percentage of the total N does 134 parents represent? Without knowing the potential total, it is hard to understand the proportion the participating parents represented and the extent of generalizing the findings.

• Do you have demographic data that show how many of the 134 parents had more than one child with autism?

Data Analysis

Phase 1: online questionnaire

• IRA-Did the author and researcher both independently code 100% of the responses?

• What training was provided to the researcher in coding responses? Did the author have background/training in coding?

Results

Parent-identified factors

• These comments are interesting; however, there does not seem to be a methodology associated with what comments were considered important, nor a method for identifying themes. Without a confirmed approach for coding comments, it is not clear exactly how these comments impact decisions, given that you cannot feasibly provide all comments. Thus, using focus group coding procedures that can provide themes is the most research validated method for analyzing comments. I would highly recommend you do this in order to answer your second research question.

Discussion

• It might make sense to organize your discussion around how it answered your two research questions. The first paragraph within the Discussion section appears to do that, but in a vague manner.

• Research in the Community identifies anxiety receiving the highest ranking in Q-sort. It is unclear how this is a community research need-could the authors clarify it? Is the research that is needed in this area related to examining how community mental health providers understand co-morbidity of conditions with autism, specifically anxiety and what treatment methods are most effective at reducing anxiety symptoms in the autism population? Or is it how anxiety impacts individuals with autism in the community?

Limitations

• The author did a nice job of explaining how the different methodologies might be a limitation. I suggest that the authors may want to further address that there seems to be limited correlation between the two methods and the outcomes. Indeed, the only use of the online questionnaire appeared to be a framework for the Q-sort. Table 3 highlights this disconnect. For example, parent, sibling, child, and family stress in the home setting was ranked near the bottom (13/15) in the online questionnaire but was ranked #1 in the Q-sort across all and primary school participants, and #2 by secondary participants. This trend is seen across the settings. There is mention of the disconnect and possible reasons, but it may also be a flawed methodology. Might the research have been more powerful if different techniques were used or if the online questionnaire might have provided more structure that would enhance more careful thought processes when parents stated priorities?

• I strongly believe you have a major limitation in answering your second research question as you did not use focus group analysis for coding the responses parents gave about the factors impacting their selections in Q-sort. You basically have not answered that question satisfactorily.

• Another limitation that may be worthy of mention is the lack of diversity of your participants. You primarily had females (mothers). Would the results have been different from a male/father perspective?

Implications

• Future research areas are vaguely described. It would help future researchers to have more specificity in what type of research should be conducted. First, replication might be an avenue to suggest. Given that your primary findings were from 9 parents only, the results are not very generalizable. Second, tightening your methodology, specifically the focus group or questions following the Q-sort.

6. PLOS authors have the option to publish the peer review history of their article (what does this mean?). If published, this will include your full peer review and any attached files.

Reviewer #1: No

Reviewer #2: No

---

## [Author Response · Author response to Decision Letter 0]

27 May 2020

Comments to the Author

Upon my review, I recommend the manuscript titled: “Listening to parents to understand their priorities for autism research” for a decision of Major Revision for publication in PLOS ONE. Overall, I find that this manuscript offers a strong contribution to the larger area of Autism research. It provides insight into the process of involving parent’s perspectives in research priority setting as well as an analysis of what some of these priorities are within the Autism community. I do however think that the arguments and organization of the manuscript need to be made stronger. Particularly, the overall justification and set up of the manuscript are unclear. While the results and discussion of the data are strong, I found myself coming to my own conclusion about the importance of this work, whereas the manuscript should have made this clearer from the start. I also think that the balance of a discussion of the process of involving parent’s perspectives and the actual data collected in for the manuscript should be considered further. I do find this work to be important to the field and important to share. I have provided some detailed comments below. The comments are in approximate chronological order in relation to the manuscript. I hope you find these comments helpful. 

Thank-you for taking the time to review our manuscript and for the helpful comments you have provided. We have provided a response to each comment below. Changes within the manuscript are highlighted with orange text for ease of reference. 

Reviewer Comments

-There are some inconsistencies in editing of the manuscript. For example, there seems to be both APA and MLA citations included in text on page 1, some paragraphs not being indented after headings, statements in the manuscript about redaction due to blind review, etc. Please review the entire manuscript for errors such as this. 

The manuscript has been checked and all errors with formatting and referencing have been resolved. 

-The introduction of the manuscript could benefit from a deeper discussion of why engagement is important.

-The discussion of the need for engagement with priority setting is not clear. It is stated that priority setting needs to be done and then two larger studies where it was done are outlined. 

We have added a more in-depth discussion of why engagement of the autism community in autism research is important and how this relates to priority setting specifically. The following has been added to pages 3 and 4 of the manuscript:

“Researchers are beginning to understand the importance of engaging members of the autism community (including individuals on the autism spectrum, their family members, educators and allied health professionals working in the field) in setting priorities. The new era of autism research seeks to engage the autism community meaningfully, valuing the collaboration between researchers and community members so that the needs and experiences of individuals on the autism spectrum and their families can inform the research priorities and agenda for research [4]. Increasing community engagement in autism research can increase the translation of science and improve the application of findings in practice, thereby working towards reducing the research to practice gap (Pellicano, Dinsomre & Charman, 2014; Pellicano & Stears, 2011). 

Research involves many phases from the conceptualization of ideas, establishment of a project, recruitment of participants and the dissemination of findings. However, research engagement often occurs partway through, often during the recruitment phase, or following the completion of a project when findings are disseminated (Goodman & Sanders Thompson, 2017). However, by this stage of the research process, the ideas have been established, the project has been conceptualized and the study is underway or completed, with decisions often pre-determined and driven by researchers’ agendas. This leaves little room for the input from non-researchers to shape the direction of the study, and unsurprisingly, this has left community members feeling dissatisfied and even frustrated with their limited involvement in the research process (Pellicano, Dinsmore & Charman, 2014). 

However, engagement can involve participation in any stage of the research process from the conceptualisation of research agendas through to the dissemination of findings (Goodman & Sanders Thompson, 2017). Thus, it is important that researchers adopt a more inclusive research ethos and make an effort to engage the community earlier in the research process. One way of meaningfully engaging the autism community in research from the outset is through priority setting, allowing the experiences, ideas and needs of community members to influence the development of a project from the beginning.”

There is also a point made about priority setting only being done for early intervention years, but then the studies outlined are not clearly described in terms of their support of this point. It seemed as though the other studies did measure more than just early intervention years. It would be helpful to clarify these points and connect the introduction arguments more directly to the findings of the reviewed studies. 

We have clarified in the text that the work by Ghanadzade et al. (2018), Leadbitter et al. (2018), Pituch et al. (2011) and Rodeger et al. (2004) were all focused on priority setting with parents during the early intervention years. We have also added in a discussion around why focusing on these same priorities in the early years at school may not translate into the same needs in the school years. 

“These studies sought to determine parent’s priorities for early intervention, including prioritizing goals for the early intervention years in preparation for school. However, as these studies focused on younger children prior to school entry, we cannot assume that the needs and priorities identified by parents during the early intervention years will remain priorities as children transition into school. Thus, it is important that the specific priorities most relevant to school age so that research can be targeted to the areas of most importance to support children during this developmental stage. As needs can change at various stages of development, needs prioritisation should occur at different life stages so supports can be tailored to individuals on the spectrum across various stages of the lifespan [11].” 

-The introduction overall was weak in terms of how prior work was being connected to the current work. Is the reader supposed to see the current study as building on past works but with a small twist? Is the reader supposed to see the current study as replicating the past works? 

The introduction has been revised and a stronger tie have been made between prior work and the current study. This has also assisted in building a stronger argument and rationale for the current study. 

-What is the Autism CRC? Please describe for the reader.

The Autism CRC is a cooperative research initiative to improve autism research across Australia. This has been explained for the reader. 

“The Autism CRC, the world’s first cooperative autism research collaboration within Australia.”

-Overall, the introduction does not provide a strong argument for why this manuscript is being written and what the manuscript is trying to get across. It would be helpful to more directly explain to the reader what the point is. 

-Why should another priority setting paper be conducted if two large-scale ones have already been done? Did the prior work miss something? Provide more justification in the introduction.

We have made amendments to the introduction in order to clarify the rationale for the current study. We have drawn on the work of den Houting and Pellicano (2019) which indicates that autism research has failed to meet the priorities of the autism community, and thus, why priority setting is needed to improve the direction of future autism research with Australia. The following has been added onto page 5 the manuscript:

“Exploration of the autism research funding in Australian between 2008 and 2017 identified that autism research within Australia has failed to align with the priorities of the autism community (den Houting & Pellicano, 2019). Therefore, more priority setting work within Australia is needed to increase the understanding of the priorities of the autism community. This will help to guide the allocation of research funding and ensure that research is being done where it is needed, and thus, is most likely to have a real impact on the lives of individuals on the autism spectrum, and their families.”

-I am not sure that I fully buy the argument that the research priorities of parents of school-aged children with autism is a gap in the literature. The manuscript described two studies in the introduction that explored research priorities. Was it supposed to be stated that those studies specifically did not evaluate parents of school-aged children? If this is the argument it needs to be made clearer or a new argument should be made regarding the importance of this work.

Yes, the other studies that focused on priority setting in the early intervention year, prior to school age. It has now been clarified that these earlier studies did not focus on school age children and why more research is needed in this area. For example, it has been identified by Gatfield (2016), that the school-aged population (children aged 5-17 years) as the priority group for future autism research. Therefore, it is important that research identify the needs of this specific group of children and the best way to achieve this is to conduct priority setting exercises with parents of school-aged children on the autism spectrum. The following has been added to the aims on pages 7-8 of the manuscript to build a stronger argument for the focus on the school-aged years:

“This study builds on the first preliminary priority setting study within Australia by Gatfield et al. (2016). Given that the largest subgroup of individuals, those living with autism identified that school age children on the spectrum (aged between 5 and 17-years) as the priority group for future research, the current study is the first within Australia with specific focus on investigating the research priorities of parents of school-aged children with autism by asking the following research questions”

-I would like to see a brief justification as to why the 3 settings (home, school, community) were chosen.

Three contexts were chosen as this allowed us to investigate the priorities of parents cross contextually. This also allows us to be able to highlight where research may be needed to support children on the autism spectrum across multiple environments. A brief justification for the choice of the three settings has now been included on page 9 of the manuscript:

“This study collected parent priorities across the three settings (home, school and community) to obtain a cross-contextual perspective of where parents’ perceive research is needed to support children on the autism spectrum across different environments.”

-It would be helpful to have a brief explanation of the “forced sort” Q-sort task of Q-methodology within the manuscript. 

A more detailed explanation of the Q-sort task has been included on pages 9-10 of the manuscript with the addition of the instructions provided to participants prior to each of the three focus groups:

“In this activity you will be arranging the top 15 priorities gathered from parents in an online questionnaire to determine where research is needed to support school-aged children on the autism spectrum at [home, school community]. Each priority is printed on a separate card and you will be using this grid to rank your priorities from most important at this end, to least important at this end. You will have 10 minutes to work independently and arrange your priorities. Once you are happy with your ranking you will use the blue-tak to stick each priority onto their respective squares on this grid”.

-Nice description of the content analysis.

Thank-you

-The quotes from the families are very helpful in the discussion of why certain priorities ranked where they did during the Q-sort method.

Thank you, we agree that the addition of the quotes obtained from the parents helped to provide some insight into parents’ decision making. 

-The parent discussion piece is very strong.

Thank-you

-Should there be any discussion about the location of the parents in terms of their responses? For example, are their responses different because they are all from Country X and the social systems in that country may be quite different from other countries, and therefore priorities may be different? Perhaps this is a point for discussion or limitations.

The majority of parents resided within Australia, with the addition of four international parents. There is the possibility that priorities may differ depending on the country and associated cultural values. 

However, the work of Gatfield et al (2016) identified that the key priority themes identified in this current study are similar to the priorities that have been identified in the large-scale international studies suggesting that some of the key priorities for parents may infact be universal. Nonetheless, this has been acknowledged as a possibility in the limitations section of the manuscript, with the addition of the following on page 24:

“It is important to acknowledge the lack of diversity in the sample with the majority of parents (97%) residing within Australia. While the priorities identified in the first Australian study by Gatfield et al. (2016) did align with those from overseas, there is the possibility that the parent priorities may be influenced by cultural and social systems within a country. Therefore, future research would benefit from comparison of priorities of parents or stakeholders across different countries”.

-It may be helpful to include a discussion on the broadness vs. narrowness of the identified priorities. Some seem to be quite broad like health and well-being while others are very specific like meltdowns and tantrums. In making suggestions about future research on these priorities, it would be helpful to consider the scope and difficulty in addressing them.

It could be assumed that a narrow term reflects a higher priority because that exact term was used by many parents, and that a broader term reflects a lower priority because it was used to reflect a number of different terms within that category. However, this is going beyond the data, because it could be that those priorities with a narrow term (i.e., tantrums/meltdowns) have a clearer commonly used term to refer to a specific behaviour. In contrast, health and well-being encapsulates a range of terms that were provided by parents to describe different states experienced by their child (i.e., illness, sleep, pain). We have added a section in to the future directions to suggest that future research could probe parents for additional clarification would be better placed to determine which priority is more pressing.

-The discussion about the cognitive processes involved in the two methods is appreciated. 

-I would like to see a more specific call to action in the discussion of the manuscript for research that is in the priority areas identified. The data from this study has clearly identified some gaps in the current research and those gaps should be highlighted in the current manuscript as a way to guide future work.

We have generated further discussion around the importance of future research in some of the key priority areas identified. We have specifically focused this discussion around three key priorities highlighted in both the current study and the other Australian priority setting study by Gatfield et al (2016). 

The following recommendations for more autism and anxiety research have been included on page 22 of the manuscript: 

“With emerging knowledge of how anxiety is impacting the lives of children on the autism spectrum, combined with the identification of anxiety as a leading priority in the work by Gatfield at al. (2016) and the current study, highlights the need for further work to understand the presentation and support the experience of anxiety in school-aged children on the autism spectrum across settings.”

The following paragraphs have been incorporated onto page 22 of the manuscript to discuss the need for more research to support the transition towards an inclusive school environment for students on the spectrum:

“As many as 70% of children on the spectrum are now attending mainstream classrooms (Clark, Vinen, Barbaro & Dissanayake, 2018), with an estimated 80% of these students requiring additional support in their school setting (Australian Government, 2018). With inclusive education a priority for parents in the current study and ‘educating’ the second highest priority themes by Gatfield et al. (2016), understandably, parents want to see research that prioritises inclusive education, with concerns that their children are not being supported within their school environment: “often there is a societal drive to place children in mainstream schools, even though the system isn’t supporting the approach”.

“Only one Australian study to date reports on the school-based provisions of students on the autism spectrum (Clark, Adams, Roberts and Westerveld, 2018), resulting in a limited understanding of what inclusive education looks like, and specifically how teachers are educating and supporting students on the autism spectrum in the classroom environment. Therefore, it is evident that more research is needed to respond to parents’ concerns surrounding inclusive education, to understand how to build a supportive inclusive environment that can best suit the learning needs and maximise the well-being of students on the autism spectrum.” 

The following has been included on page 20 of the manuscript to discuss the need for research to inform professional development opportunities for teachers, with teacher/staff education and support identified as a top priority by parents. 

“Research needs to first explore teachers’ preferences for autism specific professional development that can inform targeted learning that can equip staff educating students on the autism spectrum.”

-The manuscript seems to err on the side of discussion of the importance of the process of gathering parents priorities and not as much on what was actually found in the study. There is importance to both and potentially more importance in the latter. I would encourage the authors to consider this balance in the overall presentation of the manuscript. 

We have amended the discussion and feel that further discussion of some of the key priorities and the need for more research to address these priorities has provided a more balance view of the priorities identified as well as the importance of engaging parents in the research process. 

Reviewer #2: Thank you for the opportunity to review the manuscript titled Listening to parents to understand their priorities for autism research. Priority research is vital to determine the targets most important to the consumers who would benefit most from the research. Considering that most grants awarded focus on biological factors, even though that does not appear to be the most desired area of research for consumers, studies similar to what the authors did in this manuscript need to occur more frequently and should be more widely disseminated. In doing this, we may be able to change the trajectory of autism research. The manuscript was well written and has potential of being a valuable contribution to the field. I have the following suggestions, comments, and questions

Thank-you for taking the time to review our manuscript exploring the priority setting of parents of children on the autism spectrum across two phases. We have responded to each of your comments and questions in turn below and feel that the changes made have improved the overall quality of the manuscript. 

Introduction

• Page 4-description of respondents to UK survey-use of word “autistic individuals”; in some circles, this is preferred while in others, person-first language is required for publication. Not sure of the PONE guidelines. This use of “autistic people, etc.” is seen in other sections of the manuscript along with person-first language; thus, its use is inconsistent.

The terminology throughout has been revised to “on the spectrum” or “on the autism spectrum” in concordance with preferences of individuals on the autism spectrum, their parents and professionals working in this area (Kenny et al. 2016). The only exception to this is the discussion of the results from the UK-survey which use the term “autistic individuals”. 

• Page 5-the last sentence before the Current Study section appears to be incomplete.

This reads incomplete because of the Vancouver referencing style but finishes with the authors. This has been modified slightly to clarify this: 

“For example, although anxiety is recognized as one of the most common co-occurring conditions for children on the spectrum [15], a systematic review on anxiety in children on the autism spectrum notes a lack of research in this area [16] while the lack of research exploring the impact of anxiety on the quality of life for children on the spectrum was discussed in a subsequent study that explored the relationship between anxiety and quality of life in children on the autism spectrum [17].”

Method

Procedure

• Were both phases using qualitative procedures? The questionnaire, as described later in the analysis section, appeared to be open-ended. Yes, both phases were qualitative, with the questionnaire open ended.

• Would Q-sort be a “focus group”? Q-sort appears to be an approach to capture both qualitative and quantitative information whereas focus group activities is a separate approach that relies on facilitation of rich discourse between participants in a group with analysis primarily arriving at a coding schema to identify themes. It may be more accurate to state Q-sort approach was used with a subset of the participants who received the questionnaire. The participants in the Q-sort did not appear to have a discussion; rather, they primarily sorted cards that reflected their ideas about priorities. It also may be more accurate to describe it as Q methodology. Although you had discussions with parents after the Q sort was completed, the coding and primary analysis seemed to focus on the Q sort and comments related to priorities. That is, there did not seem to be a qualitative research plan for identifying themes for the parent-identified factors that influenced their decisions in the Q-sort.

We have modified the method of ‘phase 2’ to focus on Q-sort methodology rather than on focus groups:

“Q-sort methodology was implemented in the second phase of the study to gain further insight into parents’ research priorities that had been established from the online questionnaire in phase 1”

Questionnaire

• Please provide more information about the on-line questionnaire. How many questions were included? Or did the questionnaire only include the one question about priorities? I am assuming that it included only one question; however, it is not clear if that is the case.

The questionnaire was advertised as, and focused on, establishing parents’ priorities for research to support their school-aged child on the autism spectrum across the home, school and community settings. The questionnaire began with a series of question about priorities for the home, school and community and then moved on to gather demographic questions about the parent and child (reported in Table 1). 

Q Sort

• How was the Q-sort facilitated? There were 3 groups, each consisting of 3 parents. Describe how the Q-sort approach was conducted for each group and how you ensured consistency of facilitation across all 3 groups.

The following instructions have been added to explain the Q-sort approach. The same instructions were delivered to parents in all three groups to maximise consistency. 

“In this activity you will be arranging the top 15 parents’ priorities for research to support school-aged children on the autism spectrum at [home, school community]. Each priority is printed on a separate card and you will be using this grid to rank your priorities from most important at this end, to least important at this end. You will have 10 minutes to work independently and arrange your priorities. Once you are happy with your ranking you will use the blue-tak to stick each priority onto their respective squares on this grid”.

Participants

• Do you know the total N of parents with children who have autism in the country/area from which you recruited? That is, what percentage of the total N does 134 parents represent? Without knowing the potential total, it is hard to understand the proportion the participating parents represented and the extent of generalizing the findings.

• Do you have demographic data that show how many of the 134 parents had more than one child with autism?

It is difficult to determine how representative the sample was. While the majority of parents were from across Australia, we cannot base this on Australian data alone as four of the parents that completed the questionnaire resided outside of Australia. However, a number of previously published studies of priority setting have had numbers below 100. We acknowledge the modest sample size in the limitations as per below

“Previous priority setting studies have reported on samples as small as 43 (Shattuck et al., 2018) and 62 (Vasa et al., 2018). However, it is acknowledged that the sample size of the online sample was modest and as the Q-sort was only conducted with 9 parents (eight mothers and one father), it is difficult to generalize the findings; an important element to consider when conducting future priority setting work. Nonetheless, this is an important first step in seeking to understand the priorities of parents of school-aged children on the spectrum.”

Data Analysis

Phase 1: online questionnaire

• IRA-Did the author and researcher both independently code 100% of the responses?

The first author independently coded 100% of the responses and reliability was completed by the second author. The following has been added into the phase 1 data analysis section provide detail on the inter-rater reliability process. 

“The second author independently coded 20% of the responses across each setting that were selected at random into categories to assess reliability.”

• What training was provided to the researcher in coding responses? Did the author have background/training in coding?

The first author was trained in content analysis upon joining the research team of the second author. Both researchers have published multiple publications using content analysis, including two together. Clark & Adams (2020). The self-identified positive attributes and favourite activities of children on the autism spectrum. Research in Autism Spectrum Disorders. Clark & Adams (2020). Parent perspectives of what helps and hinders their child on the autism spectrum to manage their anxiety. Clinical Psychologist. 

Results

Parent-identified factors

• These comments are interesting; however, there does not seem to be a methodology associated with what comments were considered important, nor a method for identifying themes. Without a confirmed approach for coding comments, it is not clear exactly how these comments impact decisions, given that you cannot feasibly provide all comments. Thus, using focus group coding procedures that can provide themes is the most research validated method for analyzing comments. I would highly recommend you do this in order to answer your second research question.

The focus on the current study was to begin to identify the priorities of parents of school-aged children on the autism spectrum, and then to have parents arrange the priorities from most to least important using Q-sort methodology. We feel the discussion amongst parents was also important to include to begin to understand their decision making. However, as we had not coded their comments into themes, we were not attempting to determine what factors impacted their decisions as this was beyond the scope of the current study. We have however taken this onboard and addressed this as a limitation and a consideration for future research. See response to limitations below. 

Discussion

• It might make sense to organize your discussion around how it answered your two research questions. The first paragraph within the Discussion section appears to do that, but in a vague manner.

We have aligned the order of the discussion with the home, school and community aspects of the research question and results. By doing this, it allows us to have clear sections to discuss the priorities for each area and how they link to the existing literature. We have made a number of amendments to support a clearer flow through the discussion which are detailed elsewhere in this response. 

• Research in the Community identifies anxiety receiving the highest ranking in Q-sort. It is unclear how this is a community research need-could the authors clarify it? Is the research that is needed in this area related to examining how community mental health providers understand co-morbidity of conditions with autism, specifically anxiety and what treatment methods are most effective at reducing anxiety symptoms in the autism population? Or is it how anxiety impacts individuals with autism in the community?

The question asked parents about research which would help to support their child when they are out in the community (e.g. at clubs, societies, supermarket) rather than help the the group of people called the community. We have clarified this in the headings of the results and discussion. 

Parents responses indicated that anxiety related to two main aspects within the community 1: anxiety impacting how their child integrated/participated in the community and 2: research that helps support the anxiety associate with integrating into the community. This has been discussed further on pages 19-20 of the manuscript:

“Anxiety received the highest overall Q-sort ranking in the community with “social anxiety” the most common response suggesting that anxiety was impacting children’s ability to interact with others in the community. One parent elaborated on the impact of their child’s anxiety on completing necessary tasks in the community such as grocery shopping “It is difficult for us to be in a busy crowded supermarket… sometimes I have had to leave without finishing my shopping” and “having to wait in ques at the shops is too difficult for my child and he becomes anxious”. In a subsequent study, parents of school-aged children on the spectrum identified that ‘noise, crowds and overstimulation’ as the leading barrier hindering children from managing their anxiety when in the community (Clark & Adams, in press) suggesting that certain elements in the community are exacerbating anxiety in children on the spectrum”. 

Limitations

• The author did a nice job of explaining how the different methodologies might be a limitation. I suggest that the authors may want to further address that there seems to be limited correlation between the two methods and the outcomes. Indeed, the only use of the online questionnaire appeared to be a framework for the Q-sort. Table 3 highlights this disconnect. For example, parent, sibling, child, and family stress in the home setting was ranked near the bottom (13/15) in the online questionnaire but was ranked #1 in the Q-sort across all and primary school participants, and #2 by secondary participants. This trend is seen across the settings. There is mention of the disconnect and possible reasons, but it may also be a flawed methodology. Might the research have been more powerful if different techniques were used or if the online questionnaire might have provided more structure that would enhance more careful thought processes when parents stated priorities?

Priority setting questionnaires often use a “top down” approach where researchers ask parents to respond to a pre-set list of priorities. Whilst this provides more structure and constraints around the questions provided to parents (so makes analysis clearer and easier), it is not truly parent-led as the researchers have determined the responses. In the current study we made the deliberate decision to take a “bottom up” approach and provide parents with an open-ended questionnaire to capture parents the entire range of parent research priorities without any constraints or restrictions of a pre-determined list of items that may prime parents’ responses. 

We acknowledge that every methodology has its strengths and limitations. While a strength of this study was the opportunity to capture parents the entire range of priorities, a limitation of doing so is less structure and prompts provided to parents. One thing that was highlighted in the current study was that method does really matter as in this case, each method, online questionnaire and q-sort provided quite different results in terms of the rankings of priorities. Therefore, it is important to consider the current findings within the context of the methods used. Future priority setting research should also be mindful of which methods are being used and the potential to produce a variation in results. This has been addressed as a limitation on page 25 of the manuscript with the following discussion of how more structure could be provided to parents with future replication:

“Future research using open-ended questionnaire formats may benefit from providing parents with some prompting or structure to guide how to approach the open nature questions without influencing their responses. For example, providing parents with the following instructions to set up the questions may provide them with more structure “What questions do you think research should try to answer in order to support children on the autism spectrum in the following settings? (These questions could begin with what does…? Why do….? How can we….? When is….? Or many other combinations)” Further, encouraging parents to provide a brief explanation or example alongside their response may have yielded more context for one-word responses such as ‘anxiety’ that could have been interpreted in a number of ways (i.e., understanding the presentation of anxiety, diagnosing, or supporting anxiety).” 

• I strongly believe you have a major limitation in answering your second research question as you did not use focus group analysis for coding the responses parents gave about the factors impacting their selections in Q-sort. You basically have not answered that question satisfactorily.

Yes, this study has only begun to explore the factors that influenced parents 

We have added in the following to the limitation section 

“While this study began to explore parents’ perceptions of why certain priorities were more important than others and has included some of the discussion from parents, this was limited to anecdotal comments from parents and did not involve formal analysis of their responses. Future replication of the Q-sort methodology with a larger and more diverse group of parents (including more fathers) would benefit from use of focus group analysis for coding parent responses about what influenced parents’ decision during the Q-sort activity.”

• Another limitation that may be worthy of mention is the lack of diversity of your participants. You primarily had females (mothers). Would the results have been different from a male/father perspective?

The lack of diversity in both samples (for the online questionnaire and focus groups) has been acknowledged in the limitation. As mother-father agreement has not been explored in priority setting for autism, is unclear whether inclusion of more fathers would have resulted in different priorities, but this is important to consider. The following has been included on page 22 of the manuscript:

“Mothers formed the large majority of respondents in both phases of this study. It is not unusual for research to have a larger representation of mothers, than fathers, with others often acting as proxy respondents for fathers with work commitments (Mitchell et al. 2007). Despite the challenges associated with recruiting and retaining fathers in research Mitchell et al. discuss the value that fathers can add to a study. It is unclear whether the priorities identified in the online questionnaire would have remained similar or differed with the inclusion of more fathers and future priority setting studies should seek to recruit a more diverse sample of parents, with more fathers where possible. Further, as the focus groups were only conducted with 9 parents (eight mothers and one father), it is difficult to generalize the findings and is important to consider when conducting future priority setting work. Nonetheless, this is an important first step in seeking to understand the priorities of parents of school-aged children on the spectrum.”

Implications

• Future research areas are vaguely described. It would help future researchers to have more specificity in what type of research should be conducted. First, replication might be an avenue to suggest. Given that your primary findings were from 9 parents only, the results are not very generalizable. Second, tightening your methodology, specifically the focus group or questions following the Q-sort.

Future research areas have been explained in more detail in response to the limitations of participants, methodology and lack of diversity in the group. We have addressed the other comments in our revisions above.

---

## [Decision Letter · Decision Letter 1]

21 Jul 2020

PONE-D-20-01529R1

Listening to parents to understand their priorities for autism research

PLOS ONE

Dear Dr. Clark,

Thank you for submitting your manuscript to PLOS ONE. After careful consideration, you will see that both reviewers were extremely positive about your revised manuscript, as am I. One minor suggestion was made regarding the description of the Q-sort procedure. Please consider whether this additional clarification is needed. Given that both reviewers have recommended acceptance of your manuscript, I do not foresee needing to send this out for another round of reviews.

We look forward to receiving your revised manuscript.

Kind regards,

Eric J. Moody, Ph.D.

Academic Editor

PLOS ONE

Reviewers' comments:

Reviewer's Responses to Questions

**Comments to the Author**

1. If the authors have adequately addressed your comments raised in a previous round of review and you feel that this manuscript is now acceptable for publication, you may indicate that here to bypass the “Comments to the Author” section, enter your conflict of interest statement in the “Confidential to Editor” section, and submit your "Accept" recommendation.

Reviewer #1: All comments have been addressed

Reviewer #2: (No Response)

2. Is the manuscript technically sound, and do the data support the conclusions?

Reviewer #1: Yes

Reviewer #2: Yes

3. Has the statistical analysis been performed appropriately and rigorously? 

Reviewer #1: Yes

Reviewer #2: Yes

4. Have the authors made all data underlying the findings in their manuscript fully available?

Reviewer #1: Yes

Reviewer #2: Yes

5. Is the manuscript presented in an intelligible fashion and written in standard English?

Reviewer #1: Yes

Reviewer #2: Yes

6. Review Comments to the Author

Reviewer #1: Comments to the Author – Revision 1

Upon my re-review, I recommend the manuscript titled: “Listening to parents to understand their priorities for autism research” for a decision of Accept for publication in PLOS ONE. I found the authors responses to my comments to be adequate and the manuscript to be better for it. Particularly, I find the arguments to be more well-rounded and the overall manuscript to have a much better balance of the discussion of the current study’s findings and the importance of the process of taking parents priorities into account. I think that what has been added to the manuscript has helped create a clearer description of the objectives of the manuscript and now provides a clearer call to action for readers. Overall this manuscript offers a great contribution to the field and I hope the authors are able to share their work here. Thank you for your continued work.

Reviewer #2: Thank you for the opportunity to conduct a second review of the revised and resubmitted manuscript titled Listening to parents to understand their priorities for autism research. I found that the authors were very responsive to the queries and comments from the first reviewers. The revised manuscript is much clearer and has potential of being a valuable contribution to the field. I have only one suggestion.

Method:

Phase 2: Q-sort. In the second paragraph, the authors stated that the top categories of priorities were individually printed onto cards. Although the authors states in the previous paragraph that Q-sort was conducted to gain further insight into the research priorates from the on-line questionnaire, and also include it in the instructions provided to the participants, it would be clearer for readers to add to this statement that the 15/13 categories of priorities were culled from the response to the on-line questionnaire.

7. PLOS authors have the option to publish the peer review history of their article (what does this mean?). If published, this will include your full peer review and any attached files.

Reviewer #1: No

Reviewer #2: No

---

## [Author Response · Author response to Decision Letter 1]

23 Jul 2020

Dear Editor,

Thank-you for providing the feedback from the most recent review of the revised manuscript entitled “Listening to parents to understand their priorities for autism research”.

In response to the request from reviewer two, we have made the following minor adjustment to the wording of the method on page 9 to clarify that all categories for home, school and community were derived from the online questionnaire.

“The top 15 categories of priorities for home and school, and all 13 research priority categories for the community were derived from the online questionnaire. Each priority was individually printed onto cards for parents to arrange from most important (+4) through to least important (-4) using the Q-sort grid (see Figure 1).”

We look forward to hearing from you

Kind regards

Dr Megan Clark

---

## [Editor Report · Decision Letter 2]

27 Jul 2020

Listening to parents to understand their priorities for autism research

PONE-D-20-01529R2

Dear Dr. Clark,

We’re pleased to inform you that your manuscript has been judged scientifically suitable for publication and will be formally accepted for publication once it meets all outstanding technical requirements.

Kind regards,

Eric J. Moody, Ph.D.

Academic Editor

PLOS ONE
---

## [Editor Report · Acceptance letter]

29 Jul 2020

PONE-D-20-01529R2 

Listening to parents to understand their priorities for autism research 

Dear Dr. Clark:

I'm pleased to inform you that your manuscript has been deemed suitable for publication in PLOS ONE. Congratulations! Your manuscript is now with our production department. 

Kind regards, 

on behalf of

Dr. Eric J. Moody 

Academic Editor

PLOS ONE